# Spatial sequestration and detoxification of Huntingtin by the ribosome quality control complex

Junsheng Yang[1], Xinxin Hao[1], Xiuling Cao[1], Beidong Liu[1]*, Thomas Nyström[2]*

[1]Department of Chemistry and Molecular Biology, University of Gothenburg, Göteborg, Sweden; [2]Institute of Biomedicine, Sahlgrenska Academy, University of Gothenburg, Göteborg, Sweden

**Abstract** Huntington disease (HD) is a neurological disorder caused by polyglutamine expansions in mutated Huntingtin (mHtt) proteins, rendering them prone to form inclusion bodies (IB). We report that in yeast, such IB formation is a factor-dependent process subjected to age-related decline. A genome-wide, high-content imaging approach, identified the E3 ubiquitin ligase, Ltn1 of the ribosome quality control complex (RQC) as a key factor required for IB formation, ubiquitination, and detoxification of model mHtt. The failure of *ltn1Δ* cells to manage mHtt was traced to another RQC component, Tae2, and inappropriate control of heat shock transcription factor, Hsf1, activity. Moreover, super-resolution microscopy revealed that mHtt toxicity in RQC-deficient cells was accompanied by multiple mHtt aggregates altering actin cytoskeletal structures and retarding endocytosis. The data demonstrates that spatial sequestration of mHtt into IBs is policed by the RQC-Hsf1 regulatory system and that such compartmentalization, rather than ubiquitination, is key to mHtt detoxification.

*For correspondence: beidong. liu@cmb.gu.se (BL); thomas. nystrom@cmb.gu.se (TN)

**Competing interests:** The authors declare that no competing interests exist.

## Introduction

The Huntington disease (HD) is predominantly inherited, with a single gene, *HTT*, encoding the Huntingtin protein, at its origin (*MacDonald, 1993*). Mutated and aggregation-prone poly-glutamine-expanded (Poly (Q)) Huntingtins (mHtt) are causing HD by toxic gain-of-functions and, possibly, dominant-negative mechanisms, which are typically manifested in aged individuals (*Ross and Tabrizi, 2011*). While the formation of mHtt inclusion bodies (IBs) correlates with toxicity and disease, such formation might, in effect, be a protective response to limit proteotoxicity (*Ross and Tabrizi, 2011*; *Arrasate et al., 2004*): For example, IB formation predicts improved survival in neurons (*Arrasate et al., 2004*) and the IB-forming mHtt103QP protein (*Figure 1a*; exon-1 with 97Q repeats) are not, or only mildly, cytotoxic even when produced at high levels in young yeast cells (*Dehay and Bertolotti, 2006*; *Duennwald et al., 2006*). In contrast, when the innate proline-rich region adjacent the poly (Q) stretch of exon-1 is removed, the protein, mHtt103Q, forms multiple small, highly cytotoxic aggregates/oligomers (*Figure 1a*) (*Dehay and Bertolotti, 2006*; *Duennwald et al., 2006*; *Meriin et al., 2002*). These aggregates are associated with the actin cytoskeleton (*Song et al., 2014*) and interfere with the cytosolic ubiquitin-proteasome-system (UPS) by sequestering the Hsp40 chaperone Sis1 (*Park et al., 2013*). Chaperones, peptides, and prion-like proteins that either prevent/modify oligomer production (*Behrends et al., 2006*; *Dehay and Bertolotti, 2006*; *Krobitsch and Lindquist, 2000*; *Muchowski et al., 2000*; *Gokhale et al., 2005*) or convert small aggregates/oligomers into IBs (*Kayatekin et al., 2014*; *Wolfe et al., 2014*) can suppress the toxicity of the proline-less exon-1, suggesting that small aggregates and oligomers are likely culprits in mHtt103Q-derived toxicity (*Arrasate et al., 2004*; *Miller et al., 2011*).

**eLife digest** Huntington's disease is a neurological disease that is caused by mutations in the gene that encodes a protein called Htt. Individuals with this mutation gradually lose neurons as they age, resulting in declines in muscle coordination and mental abilities. The mutant Htt proteins tend to form clumps inside cells, but it is not clear if these clumps are the cause of the disease symptoms or whether they have a protective effect.

Yang et al. used yeast as a model to investigate whether the mutant Htt proteins need other molecules to allow them to form clumps. The experiments identified several new molecules that are required for mutated Htt to form clumps. Some of these are components of a system called the Ribosome Quality Control (RQC) complex, which monitors newly made proteins and labels abnormal ones for destruction. However, Yang et al.'s findings suggest that the RQC complex regulates the formation of Htt clumps through a different pathway involving a protein called heat shock factor 1. In this case, cells would need to fine-tune heat shock factor 1 activity to make mutant Htt proteins clump together to protect cells from damage.

Future experiments should expand Yang et al.'s findings to animal models of Huntington's disease and identify which other molecules contribute to the formation of Htt clumps. One challenge will be to find out why older neurons fail to form clumps of Htt proteins, and whether this can be overcome by drugs that boost the activity of the molecules that Yang et al. identified.

Ubiquitination is another process suggested to prevent mHtt toxicity in both mammals (*Steffan, 2004*) and yeast (*Willingham et al., 2003*). IBs of mHtt contain ubiquitin in mice (*Davies et al., 1997*) and the human ubiquitin-conjugating enzyme, hE2-25K, interacts with mHtt, which has been shown to be ubiquitinated in both humans and flies (*Kalchman et al., 1996*; *Steffan, 2004*). However, an E3 ubiquitin ligase directly responsible for mHtt ubiquitin-tagging, IB formation, and detoxification has not been identified.

## Results

We approached mHtt toxicity by a different route than recent mHtt103Q toxicity-suppression screens (*Kayatekin et al., 2014*; *Mason et al., 2013*; *Wolfe et al., 2014*) by asking if the non-toxic, IB-forming mHtt103QP carrying the innate proline-rich stretch of exon-1, requires trans-acting factors to form IBs and if such factors convert mHtt103QP into non-toxic conformers. This approach was prompted also by our finding that the ability to form large and single mHtt103QP IBs was lost upon mother cell aging and the mHtt proteins accumulated instead in multiple, three or more smaller aggregates per cell, referred to as Class 3 cells (*Figure 1b*; Class 1 cells contain one aggregate and Class 2 cells contain two aggregates). To identify trans-acting factors required for IB formation in an unbiased genome-wide manner, we used high content microscopy (HCM) and a galactose-regulated version of mHtt103QP, which we introduced into the ordered yeast deletion library (SGA-V2) (*Tong, 2001*) of *S. cerevisiae* (*Figure 1c*). Upon galactose-induction, mHtt103QP formed aggregates in about 50% of the cells within 180 min (*Figure 1d*) and 70% of these cells contain one large IB. HCM was used to identify mutants that formed multiple aggregates/oligomers rather than a big IB (Class 3 mutants; *Figure 1e*), which revealed that IB formation requires proteasome/chaperone and ubiquitination functions, Golgi-vesicle trafficking, mRNA transport/metabolism, and cell cycle control (*Figure 1f&g*, see *Supplementary file 1* for a list of confirmed mutants). Among these factors, Ltn1 and Rqc1 are especially interesting as they are both partners of the ribosome quality control complex (RQC) (*Brandman et al., 2012*) and Ltn1 is the yeast homologue of the E3 RING ubiquitin ligase Listerin of mammalian cells (*Bengtson and Joazeiro, 2010*), which reduced activity causes premature neurodegeneration in mice (*Chu et al., 2009*).

Complementation analysis revealed that the ubiquitin E3 ligase activity of Ltn1 was required for both mHtt103QP IB formation (*Figure 2a*) and ubiquitination (*Figure 2b*). It's been reported that the absence of Ltn1, but not Rqc1, results in the failure to tag non-stop protein with ubiquitin (*Brandman et al., 2012*). Contrasting such data on non-stop proteins, both Ltn1 and Rqc1-deficieny resulted in a failure of cells to tag also full-length mHtt103QP properly with ubiquitin (*Figure 2b*,

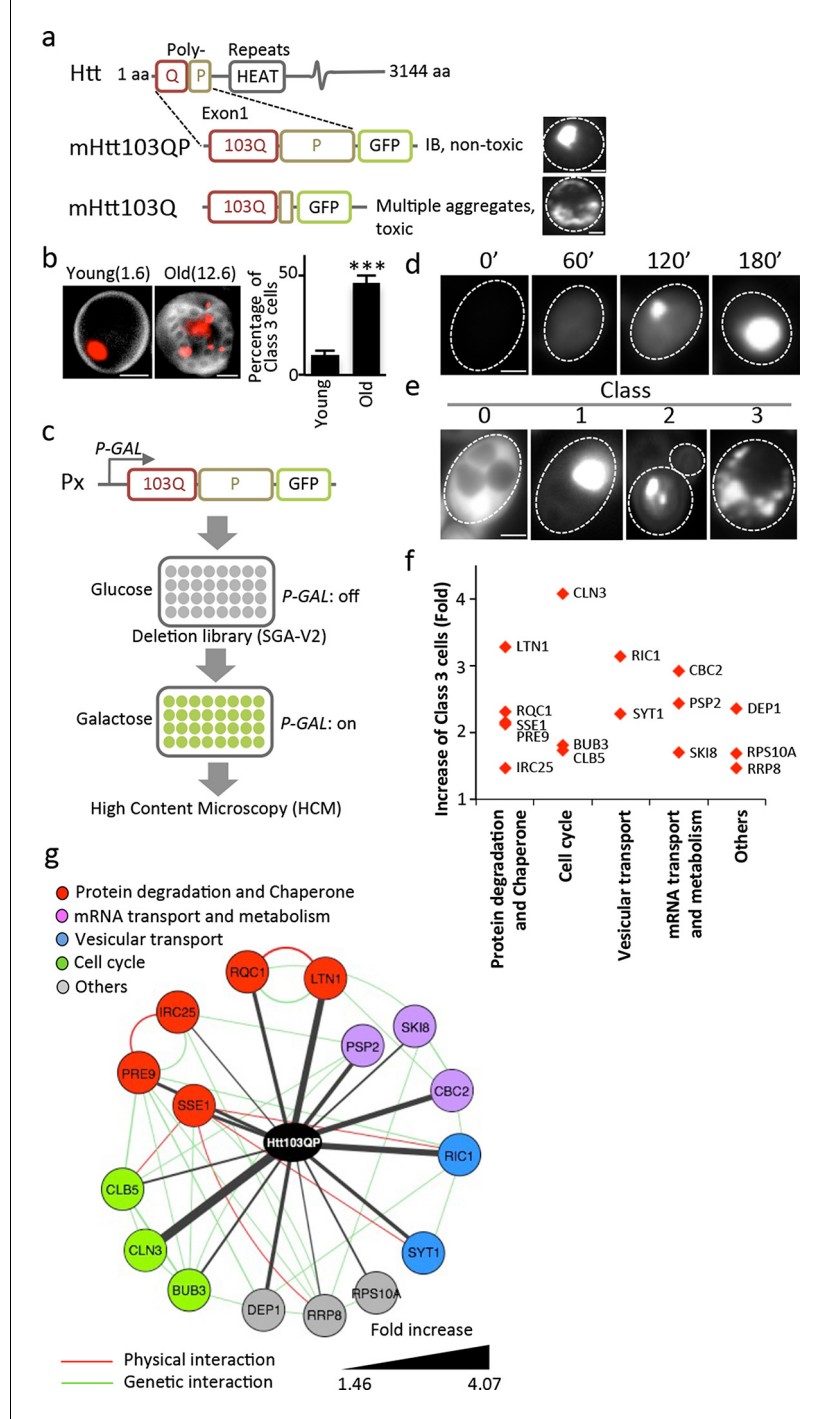

**Figure 1.** Screen approach and mHtt IB-forming mutants. (a) Aggregation of different mHtt reporters as indicated. (b) Morphology of mHtt103QP aggregates (red) in young and old (1.6 and 12.6 bud scars (white), respectively) cells. Scale=2 μm. Bar graph shows percentages of Class3 cells in young and old cells. Mean ± s.d. (c) Schematic description of the HCM-based screen. (d) Htt103QP aggregation 0, 60, 120 and 180 min after *HTT103QP* induction. (e) Representative pictures of Class 0, 1, 2 and 3 cells. (f) Mutants displaying increased% of Class 3 cells, grouped according to functions. Y-axis shows fold increase relative to wild type. (g) Physical (red) and genetic (green) interaction between Class 3 genes/proteins and their quantitative interaction (thickness of grey lines) with mHtt103QP as indicated.

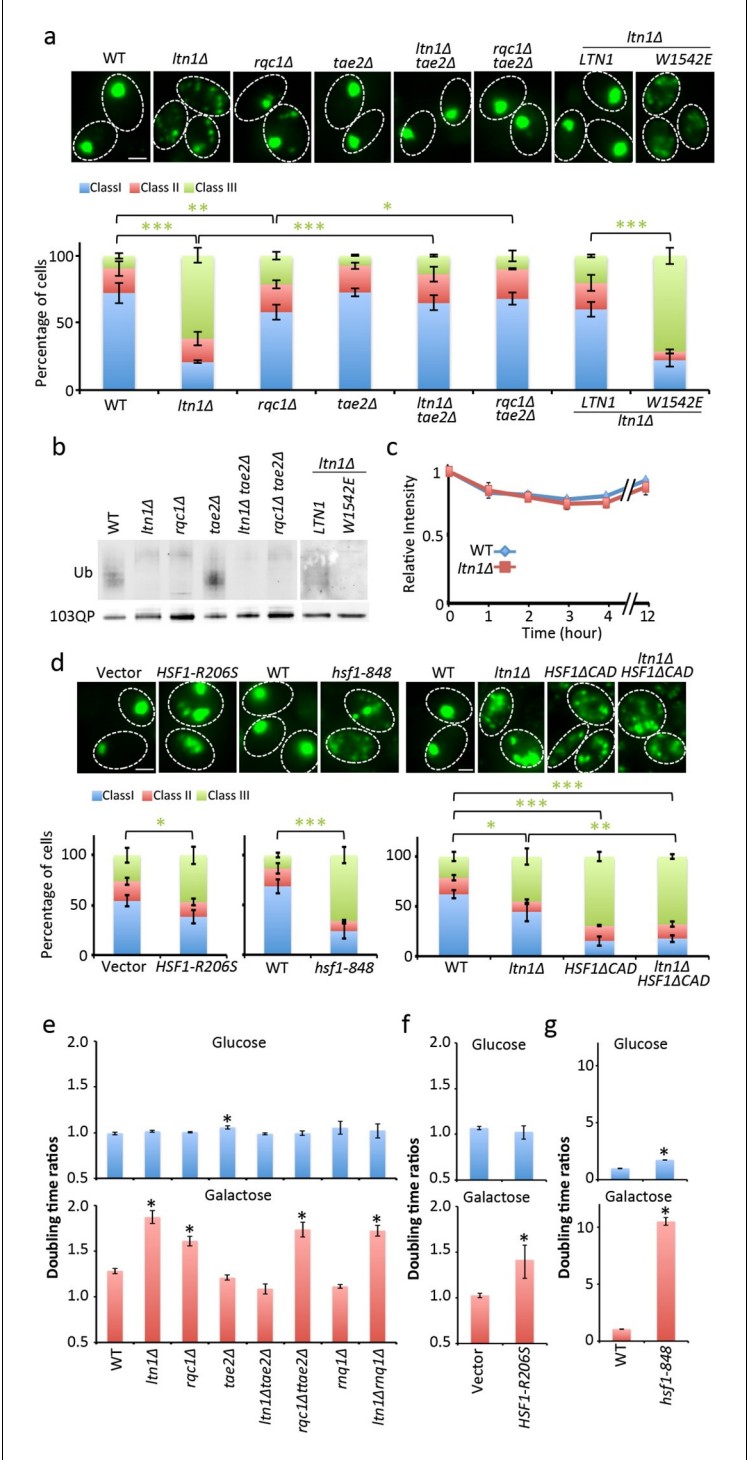

**Figure 2.** Role of RQC in mHtt103QP IB formation ubiquitination and toxicity. (a, d)Htt103QP aggregate numbers (% Class 1,2&3 cells; see *Figure 1*) in mutants as indicated. W1542E encodes a ubiquitin-ligase-defect Ltn1 protein. *HSF1-R206S* encodes a hyper-active Hsf1. The *hsf1-848* is a conditional *ts* mutant while *HSF1ΔCAD* lacks the c-terminal trans-activating domain. Scale=2 μm. Bar graphs show % of Class 1, 2 and 3 cells in each strain. Mean ± s.d. (b)Ubiquitination of mHtt103QP in strains from 'a'. (c) Htt103QP stability in WT and *ltn1Δ* cells after a block in protein synthesis. Mean ± s.d. e-g. Fitness (see Materials and methods) of strains carrying pYES2-mHtt103QP-GFP compared to pYES2-GFP. Results from Galactose (mHtt induced) and Glucose (mHtt repressed) are shown. Ratios were calculated from the mean of three repeats (error bars are 95% confidence intervals) for WT, RQC, and *rnq1Δ* mutants (e) *HSF1-R206S* (f) and *hsf1-848* (g).

*Figure 2 continued on next page*

*Figure 2 continued*

The following figure supplements are available for figure 2:

**Figure supplement 1.** Western blot of His-Ub pull-down mHtt103QP in RQC mutants.

**Figure supplement 2.** FRAP assay of mHtt103QP aggregate in Wt and RQC mutants.

**Figure supplement 3.** Ltn1-GFP co-localize with mHtt103QP-mRFP.

**Figure supplement 4.** mHtt levels chase after cycloheximide treatment.

**Figure supplement 5.** mHtt103QP aggregate in *ltn1Δtae2Δ* is also co-localized with dense actin structures.

*Figure 2—figure supplement 1*) and to form IBs, even though the effect of *rqc1Δ* was markedly smaller than *ltn1Δ* on IB formation (*Figure 2a*). Moreover, both soluble and aggregated mHtt103QP was stable in the absence and presence of Ltn1 (*Figure 2c*, *Figure 2—figure supplement 2*), and the levels of soluble and aggregated mHtt103QP was somewhat lower in *ltn1Δ* cells (*Figure 2—figure supplements 2 & 4*). These data suggest that Ltn1 is involved in mHTT103QP sequestration into IBs rather than its decay.

Ltn1-, and to a lesser extent, Rqc1-deficieny results in hyper-activation of the heat shock transcription factor Hsf1 through the RQC component Tae2 and such activation can thus be suppressed by removing the *TAE2* gene (*Brandman et al., 2012*). We found that deleting *TAE2* in *ltn1Δ* or *rqc1Δ* cells restored IB formation (*Figure 2a*, *Figure 2—figure supplements 2 & 5*) but did not restore ubiquitination (*Figure 2b*), demonstrating that ubiquitination is not an absolute requirement for the formation of mHtt103QP IBs. Moreover, overproducing a hyperactive Hsf1 (*HSF1-R206S* [*Hou et al., 2013*]) alone was sufficient to reduce IB formation, as was reducing Hsf1 activity using the *hsf1-848* (ts) allele (*Figure 2d*). demonstrating that maintaining a proper, intermediate, range of Hsf1 activity is required to efficiently sequester mHtt103QP into IBs. In support of this notion, a deletion in the C-terminal trans-activation domain of Hsf1 resulted in defects in IB formation that could not be further abrogated by an *ltn1* deletion (*Figure 2d*).

The mHtt103QP protein displays no obvious toxicity in yeast (*Dehay and Bertolotti, 2006*; *Duennwald et al., 2006*) but we found that it became detrimental in the absence of Ltn1, and to a somewhat lesser extent, Rqc1 (*Figure 2e*), supporting the idea that IB formation protects the cell against Huntingtin toxicity. Consistently, a *tae2Δ* mutation completely suppressed the toxicity of mHtt103QP in the *ltn1Δ* cells (*Figure 2e*). Since the *TAE2* deletion did not restore mHtt103QP ubiquitination, we conclude that IB formation is more important than ubiquitination for the detoxification of mHtt103QP, at least in the yeast model system. Contrasting the *LTN1* data, the absence of *TAE2* failed to fully suppress toxicity in *rqc1Δ* cells indicating that the roles of Ltn1 and Rqc1 in RQC are overlapping (*Brandman et al., 2012*) but not identical. Consistent with small mHtt103QP aggregates/conformers being toxic, both overactive and diminished Hsf1 activity rendered mHtt103QP toxic (*Figure 2f&g*). Since the proline-less, intrinsically noxious, mHtt103Q protein requires the presence of the prion-forming protein Rnq1 to display cytotoxicity in yeast (*Meriin et al., 2002*), we tested whether the toxicity of mHtt103QP in Ltn1-deficient also required the presence of Rnq1 and found that this was not the case (*Figure 2e*).

The small cytotoxic mHtt103Q aggregates have been shown to associate with the actin cytoskeleton (*Song et al., 2014*), and we, therefore, investigated if mHtt103QP in wild type and *ltn1Δ* cells likewise interacted with and affected actin cytoskeletal structures. First, using co-staining with the misfolded protein Ubc9ts-mCherry, we confirmed that the mHtt103QP proteins of wild type cells were deposited in IBs adjacent to the Ubc9ts-associated insoluble-protein-deposit, IPOD (*Kaganovich et al., 2008*) (*Figure 3a*). Super resolution, three-dimensional structured illumination microscopy (SIM) revealed that these mHtt103QP IBs were associated with dense actin cytoskeletal structures (*Figure 3b*, *Video 1*). Moreover, the actin cytoskeleton appears to harness latent mHtt103QP toxicity as a screen for conditional *ts* mutations causing synthetic sickness/lethality with mHtt103QP (*Figure 3d&e*) revealed that cells carrying *ts* mutations in genes encoding actin itself

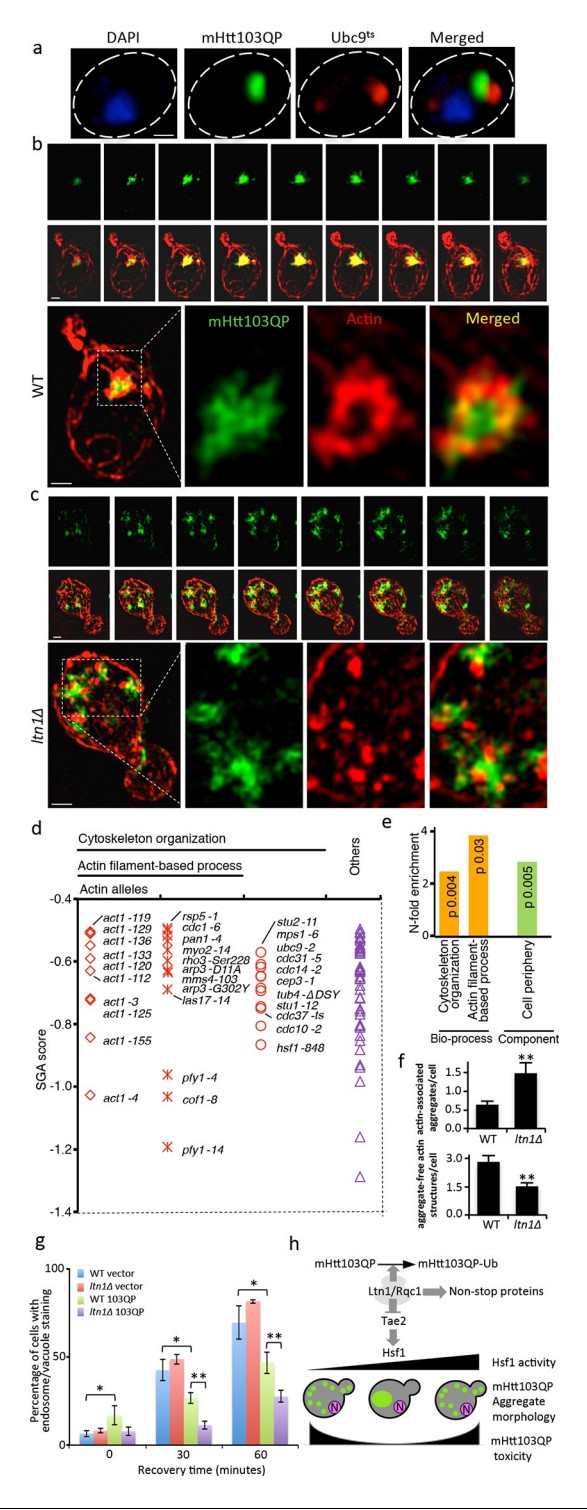

**Figure 3.** Role of actin in Htt103QP detoxification. (**a**)Co-localization of mHtt103QP IBs and UBC9[ts] IPODs. (**b, c**) Actin structures (Red; phalloidin) and mHtt103QP (Green; GFP) aggregates in WT and *ltn1Δ* analyzed by 3D-SIM. Scale=1 μm. (**d**)Essential *ts*-alleles increasing toxicity of mHtt103QP, grouped according to biological processes. (**e**) Functional enrichment analysis of mHtt103QP-sensitive *ts* mutants. (**f**) Number of actin-associated aggregates and aggregate-free actin structures in WT and *ltn1Δ* cells. Mean ± s.d. **g.** Endocytotic activity in WT and *ltn1Δ* cells analyzed by FM4-64FX uptake to vacuoles. Mean ± s.d. (**h**)A model of the regulation of mHtt103QP IB formation and toxicity by RQC components and Hsf1.

*Figure 3 continued on next page*

*Figure 3 continued*

The following figure supplements are available for figure 3:

**Figure supplement 1.** FM4-64FX stained cells.Images of FM4-64FX stained cells corresponding to *Figure 3g*.
**Figure supplement 2.** Actin integrity of Wt and a. Actin staining of Wt and *ltn1Δ* cells, b. Quantification of actin depolarization of Wt and *ltn1Δ* cells, see Materials and methods for details.

(*act1*), profiling (*pfy1*) involved in actin polymerization, cofilin (*cof1*) regulating assembly/disassembly of actin filaments, Arp3 of the actin-nucleation center, Las17, an activator of Arp2/3 and actin assembly factors, and Mss4, a phosphatidylinositol-4-phosphate 5-kinase involved in actin cytoskeleton organization, were drastically sensitized to mHtt103QP (*Figure 3d&e*, also see *Supplementary file 2* for a list of alleles). The multiple mHtt103QP aggregates formed in *ltn1Δ* cells also co-localized with actin cytoskeletal structures (*Figure 3c*, *Video 2*), akin to those of the toxic mHtt103Q aggregates reported previously (*Song et al., 2014*). Actin-mHtt103QP-associated structures were more abundant in Ltn1-deficient cells than in wild type cells whereas the number of aggregate-free forms of actin structures, including actin patches, was reduced (*Figure 3f*). Because the actin cytoskeleton is required for proper endocytosis, we tested the effect of mHtt103QP and an *ltn1* deletion on the rate of endocytic internalization of the dye FM4-64, and found that Htt103QP retarded endocytosis and that such retardation was more pronounced in cells lacking Ltn1 (*Figure 3g*; *Figure 3—figure supplement 1*). In contrast, Ltn1 deficiency did not by itself cause actin cytoskeleton defects or endocytosis retardation (*Figure 3g*, *Figure 3—figure supplement 2*).

## Discussion

The conserved Listerin (Ltn1) E3 ligase is a key factor involved in targeting protein products derived from defective mRNA or aborted translation for degradation by the 26S proteasome (*Bengtson and Joazeiro, 2010*; *Brandman et al., 2012*). Upon translation stalling, ribosome recycling factors dissociate 80S ribosome-nascent chain complexes to 60S ribosome-nascent chain-tRNA complexes, which are recognized by Ltn1 and Tae2 (*Shen et al., 2015*; *Shao et al., 2015*; *Shao et al., 2013*). Both nascent chains and, for example, K12- and R12-arrested polypeptides are substrates for Ltn1-dependent ubiquitin tagging, which signal their destruction by the 26S proteasome (*Bengtson and Joazeiro, 2010*; *Brandman et al., 2012*; *Preissler et al., 2015*). Herein, we report on another pivotal role of Ltn1 in protein quality control – detoxification of mutant Huntingtin through a Tae2/Hsf1-dependent sequestration of mHtt103QP into actin-associated inclusions (*Figure 3h*). As depicted in *Figure 3h*, the effect of Ltn1 on mHtt103QP aggregation appears to act through Tae2, which in turn is known to negatively control Hfs1 activity (*Brandman et al., 2012*). Thus, the presence of Tae2 is known to cause hyperactivation of Hsf1 when *LTN1* is deleted (*Brandman et al., 2012*), which could be enough to inhibit IB formation. On the other hand, mutations reducing Hsf1 activity also inhibited IB formation suggesting that maintaining a proper, intermediate, range of Hsf1 activity is required to

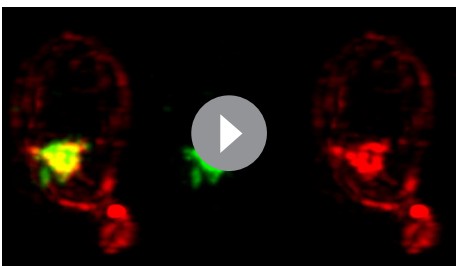

**Video 1.** 3D structures of mHtt103QP aggregate and actin in *WT*. mHtt103QP aggregates (green) and actin (red) structures of a *WT* cell shown in *Figure 3b*.

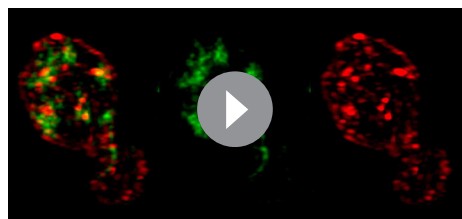

**Video 2.** 3D structures of mHtt103QP aggregate and actin in *ltn1Δ*. mHtt103QP aggregates (green) and actin (red) structures of a *ltn1Δ* cell shown in *Figure 3c*.

efficiently sequester mHtt103QP into IBs (*Figure 3h*). In worms, elevated production of small heat shock proteins through Hsf1 activity has been shown to delay the onset of polyglutamine-expansion protein aggregation (*Hsu, 2003*) and reducing *hsf-1* activity accelerates aging (*Hsu, 2003*; *Morley and Morimoto, 2004*). Reciprocally, *hsf-1* overexpression extends worm lifespan (*Hsu, 2003*). (*Baird et al., 2014*). The data presented here, however, demonstrate that both Hsf1 elevation and Hsf1 deficiency in cells expressing the Huntingtin disease protein is detrimental (*Figure 3h*), suggesting, again, that a fine balance of Hsf1 activity has to be maintained to assuage proteotoxicity. This notion might explain why alterations in Hsf1 levels in mammalian cells have been shown to either inhibit mHtt IB formation (*Fujimoto et al., 2005*) or lower the concentration threshold at which HTT forms IB (*Bersuker et al., 2013*). These results raise the question of whether age-dependent penetrance of HD could be due to a reduced Hsf1 activity in aging tissues or a malignant hyperactivation of Hsf1. The latter scenario could be the result of an age-dependent increase in translational processivity errors, which could titrate the RQC complex eliciting a Tae2-dependent activation of Hsf1 (*Figure 3h*), possibly through Tae2-directed tagging of incomplete translation products with carboxyl-terminal Ala and Thr extensions. (*Shen et al., 2015*).

The exact mechanism behind Hsf1-dependent modulation of mHtt IB formation might be complex in that Hsf1 targets other genes than heat shock genes. It has been shown in worms that over-expression of hsf-1, with or without the C-terminal trans-activation domain, elevates the levels of pat-10, a troponin-like protein, that increase actin cytoskeleton integrity leading to lifespan extension and resistance to proteotoxic stress (*Baird et al., 2014*). Thus, it is possible that Hsf1 may regulate mHtt IB formation/toxicity in the yeast model system through the regulation of actin cytoskeleton dynamics since we found that mHtt103QP is associated with dense actin structures and that genes involved in actin dynamics are required to harness the latent toxicity of mHtt103QP. In addition, our data cannot rule out the possibility that the expression of mHtt in general raises proteostasis stress in the cell leading to Hsf1 activation and that such activation is epistatically affecting the effect of Ltn1-deficieny.

## Materials and methods

### Plasmids, yeast strains, and growth conditions

Plasmids and yeast strains used in each assay and figure were specified in *Supplementary file 3A and B*.

Yeast cells were grown at 30°C if not specified, in YPD (BY4741 background), YPAD (W303 background) or corresponding synthetic drop-out media with antibiotics. For all galactose induction experiments, yeast cells were pre-cultured, diluted, and re-grown in media with 2% raffinose until mid-log phase (OD600=0.5). 2% galactose was then added to induce expression for desired time. For temperature sensitive strains (except Ubc9[ts], see below), cells were pre-cultured at 22°C and switched to 30°C during experiments.

HSF1 and HSF1ΔCAD in the W303-1A background (as described in [*Eastmond and Nelson, 2006*]) was a gift from Dr. H Nelson (University of Pennsylvania, USA). The mHtt103QP plasmid pYES2-103QP-GFP (as described in [*Meriin et al., 2007*]) was a gift from Dr. M Sherman (Boston University, USA). Plasmid pYES2-GFP (as described in [*Preval et al., 2006*]) was a gift from Dr. C Forestier (CEA, France). Plasmids pGAD-HA-Ltn1 and pGAD-HA-Ltn1-1542E (as described in [*Bengtson and Joazeiro, 2010*]) were gifts from Dr. CJoazeiro (The Scripps Research Institute, USA). Plasmid pRS416-TEF1-Hsf1M (as described in [*Hou et al., 2013*]) was a gift from Dr. J Nielsen (Chalmers University of Technology, Sweden). Plasmid pADH-His-Ub (*Lu et al., 2014*) was a gift from Dr. S Jentsch (Max Planck Insititute of Biochemistry, Germany).

### Strain and plasmid constructions

The pYES2-mHtt103QP-GFP plasmid was transformed to SGA-V2 single gene knock-out collection by a robotic SGA procedure to generate the strain collection SGA-V2-pYES2-mHtt103QP-GFP (S2Y103QPG) for HCM-based screen (*Tong, 2001*; *Tong, 2004*). A control plasmid pYES2-GFP was also transformed to SGA-V2 collection to build SGA-V2-pYES2-GFP (S2YG) collection as negative controls for toxicity assays.

*ltn1Δ::natMX4* in BY4741, W303 and W303 *HSF1ΔCAD*, SGA *rnq1Δ* backgrounds and *tae2Δ::natMX4* in SGA *rqc1Δ* and SGA *ltn1Δ* backgrounds were all generated by PCR-mediated gene deletion.

The coding sequence of URA3 in pYES2-mHtt103QP-GFP and pYES2-GFP were replaced by hphMX4 cassette via PCR-mediated gene deletion, to generate pY2H-mHtt103QP-GFP and pY2H-GFP plasmids to make them compatible with URA3 plasmids. The template used to amplify hphMX4 is plasmid pAG32 (*Goldstein and McCusker, 1999*).

### Isolation of old cells

Isolation of old cells was carried out via the biotin-streptavidin magnetic beads binding system as previously described (*Sinclair and Guarente, 1997*). Old cells ('Old' in *Figure 1b*) were labeled with EZ-Link NHS-Biotin (Thermo Fisher Scientific, Waltham, MA), first aged in glucose media for two overnights and then in raffinose media for one overnight before harvesting; young cells ('Young' in *Figure 1b*) were the progenies of the old cells generated in the last overnight culturing in raffinose media. Both young and old cells were induced for mHtt103QP-GFP expression for 3 hr and then fixed. Mean ages of samples were assessed by counting bud scars stained by Calcofluor white (Sigma-Aldrich, St. Louis, MO). Three parallel repeats were performed.

### HCM-based screen

Each strain from the S2Y103QPG collection was pre-cultured, induced for mHtt103QP-GFP expression as described earlier and fixed with 3.7% formaldehyde at room temperature for 30 min in 96-well plates. For image capturing, appropriate amount of fixed cells were transferred to new 96-well plates and imaged with the ImageXpress MICRO (Molecular Devices, Sunnyvale, CA), an automated cellular imaging system. Customized sub-program of the software MetaXpress (Molecular Devices) was applied on the obtained images for quantification. All mutants that showed statistically significant increase larger than three times the variance of the wild type were restreaked and re-tested individually and analyzed manually to confirm the phenotypic differences observed in the screen. At least 300 cells were counted in the manual confirmation.

### Microscopy

Cell images (except for 3D-SIM images in *Figure 3b&c*) were obtained via Zeiss Axio Observer.Z1 inverted microscope and Zen Pro 2012 software (Carl Zeiss AG, Germany). Filter sets used are: 38 HE GFP for mHtt103QP-GFP, 43 HE DsRed for Ubc9$^{ts}$-MCherry, 45 Texas Red for FM4-64FX and 49 DAPI for DAPI and Calcofluor white. Images in *Figure 1b* and *Figure 3a* were deconvolved by ImageJ software and plugin 'Iterative deconvolve 3D', maximum number of iterations set to 15 and 10 respectively.

IB morphology tests were performed three times for each strain in *Figure 1b*, *Figure 2a, d*; 100 cells with aggregates were analyzed and quantified for each repeat.

### Immunoprecipitation (IP) and Western blot

Whole cell protein extracts were obtained via mild alkali treatment and IPs by anti-FLAG M2 affinity gels (Sigma) were carried out following previously published protocols (*Bengtson and Joazeiro, 2010*). mHtt103QP-GFP was expressed for 3 hr in all samples. Western blotting was done as described before (*Molin et al., 2011*) using an XCell SureLock MiniCell (LifeTechnologies) and Immobilon-FL PVDF membranes (Millipore, Billerica, MA). Ubiquitination signals were detected by a rabbit polyclonal anti-ubiquitin antibody (ab19247; AbCam, United Kingdom). The mHtt103QP-GFP was detected by a chicken polyclonal anti-GFP antibody (ab13970; AbCam).

### mHtt103QP-GFP stability assay

The stability of in vivo mHtt103QP-GFP by FACS was determined by the change of GFP fluorescent signal strength after inhibition of protein synthesis by cycloheximide, as described previously (*Song et al., 2014*).

Soluble protein and protein aggregates were separated by ultracentrifugation as described in (*Song et al., 2014*) and then quantified by Western blotting. mHtt protein levels were standardized to total protein levels determined by Coomassie Brilliant Blue staining of the membrane.

## Doubling time determinations

Doubling time was determined by the Bioscreen Assays as described (*Warringer et al., 2003*), in media with either 2% glucose or 2% galactose after overnight pre-culturing in media containing 2% raffinose as the only carbon source. Three parallel replicates were run for each strain.

## Actin staining

Actin structures were stained by Alexa568-phalloidin (Thermo Fisher Scientific) as described (*Liu et al., 2010*). For quantifications in *Figure 3f,* Z-stack serial images were analyzed. To avoid possible bias caused by different distributions of cells at different cell cycle stages, only mother cells in budding events with undivided nucleus (determined by DAPI staining) were counted (*Anderson et al., 1998*).

## Localization of mHtt103QP aggregates

Both mHtt103QP-GFP and Ubc9$^{ts}$-mCherry were expressed for 3 hr at 28°C. The cells were then incubated at 37°C for 30 min to trigger Ubc9$^{ts}$ aggregate formation. Cells were fixed and washed immediately after the 37°C treatment.

## 3D-SIM microscopy

3D-SIM microscopy images were obtained as previously reported (*Song et al., 2014*).

## SGA analysis

SGA analysis of the ts-allele collection was performed and scored as previously described (*Wagih et al., 2013*; *Costanzo et al., 2010*; *Li et al., 2011*). The cut-off for the screen was -0.5 in score from the screen.

## Functional enrichment and network analysis

The functional enrichment analysis of Htt103QP essential synthetic sick interactors was based on the result from Gene Ontology Term Finder (*Boyle et al., 2004*) using the SGA ts-V5 array (787 ts alleles, covering 497 essential genes) as the background list.

Cytoscape 3.2.0 (*Saito et al., 2012*) was used for interaction network analysis of hits with increased class 3 aggregates. The physical interactions between the hits were obtained from Bio-GRID interaction database (*Breitkreutz et al., 2008*) using GeneMANIA plugin (*Warde-Farley et al., 2010*)

## Assessment of actin depolarization and endocytosis

Actin depolarization was quantified as described in (*Anderson et al., 1998*).

Endocytosis was assessed by tracking FM4-64FX (Thermo Fisher Scientific) internalization in live cells as described (*Baggett et al., 2003*) with minor modifications. Yeast cells were strained on ice for 30 min with FM4-64FX after 3 hr expression of mHtt103QP-GFP. Cells were then incubated in YPD at 30°C in dark. Z-stack images of samples taken after 0, 15, 30, 45 and 60 min incubation at 30°C were captured and analyzed.

## Flourescence recovering after photobleaching (FRAP) assay

FRAP of mHtt103-QP aggregates was carried out on LSM 700 Axio Observer.Z1 (Carl Zeiss). Images were captured every second for 90 s after photobleaching. Fluorescent intensities of the bleached region were quantified via ImageJ.

## His-Ub pull-down assay

His-Ub pull-down assay was carried out as described in (*Tansey, 2006*) with minor modifications. His-tagged Ub was expressed from pADH-His-Ub and pulled down via Dynabeads His-tag (Thermo Fisher Scientific).

## Statistics

For bar graphs in *Figure 1b*, *2a*, *2d*, *3f*, *3g*, data shown are mean of three replicates ± s.d., unpaired two-tailed t-test was used to compare mean values. Statistical significance was indicated as *p<0.05; **p<0.01; *** *P*<0.001.

For the bar graph in *Figure 2 e-g*, data shown is the ratio of means ± 95% confidence interval. The confidence intervals were calculated based on Fieller's theorem (*Fieller, 1940*) by an online-calculator http://www.graphpad.com/quickcalcs/ErrorProp1.cfm (GraphPad Software, La Jolla, CA).

## Acknowledgements

The authors would like to thank C Boone D Kaganovich, H Nelson, M Sherman, C Forestier, C Joazeiro, J Nielsen and S Jentsch for providing materials essential to this work. We thank Julia Fernandez-Rodriguez for the support on 3D-SIM microscopy, and we Acknowledge the Centre for Cellular Imaging at the Sahlgrenska Academy, University of Gothenburg for the use of imaging equipment and for the support from the staff. This work was supported by grants from the Swedish Natural Research Council (TN:VR 2010-4609) and (BL: VR 2015-04984) and the Knut and Alice Wallenberg Foundation (Wallenberg Scholar) and ERC (Advanced Grant; QualiAge) to TN, the Swedish Cancer Society (CAN 2015/406) and Stiftelsen Olle Engkvist Byggmästare Foundation and Carl Trygger Foundation (CTS 14: 295) to BL. The research leading to these results has received funding from the People Programme (Marie Curie Actions) of the European Union's Seventh Framework Programme (FP7/2007-2013) under REA grant agreement n°608743 (a mobility for regional excellence, MoRE, fellowship to BL).

## Additional information

### Funding

| Funder | Grant reference number | Author |
| --- | --- | --- |
| Vetenskapsrådet | 2010-4609 | Thomas Nyström |
| Vetenskapsrådet | 2015-04984 | Beidong Liu |
| Knut och Alice Wallenbergs Stiftelse | Wallenberg Scholar | Thomas Nyström |
| European Research Council | Advanced Grant; QualiAge | Thomas Nyström |
| Cancerfonden | CAN 2015/406 | Beidong Liu |
| Stiftelsen Olle Engkvist Byggmästare | | Beidong Liu |
| Carl Tryggers Stiftelse för Vetenskaplig Forskning | CTS 14: 295 | Beidong Liu |
| European Commission | FP7/2007-2013 | Beidong Liu |

The funders had no role in study design, data collection and interpretation, or the decision to submit the work for publication.

### Author contributions

JY, designed the study and experiments, performed old cell isolations, the HMC screen, IP, FACS and endocytosis assay, performed microscopy, toxicity assays and statistics, assisted to write the manuscript, Conception and design, Acquisition of data, Analysis and interpretation of data, Drafting or revising the article; XH, performed microscopy, toxicity assays and statistics, performed the SGA assay, Acquisition of data, Analysis and interpretation of data; XC, performed the His-Ub pulldown assay, Acquisition of data; BL, designed the study and experiments, designed the HMC screen and analyzed the HMC data, performed the SGA assay, performed 3D-SIM imaging, assisted to write the manuscript, Conception and design, Acquisition of data, Analysis and interpretation of data, Drafting or revising the article; TN, designed the study and experiments, wrote the manuscript, Conception and design, Drafting or revising the article

Author ORCIDs

Beidong Liu, http://orcid.org/0000-0001-6052-8411

Thomas Nyström, http://orcid.org/0000-0001-5489-2903

## Additional files

**Supplementary files**

• Supplementary file 1. List of confirmed mutants from the HCM-based screen that have increased Class 3 cells.

• Supplementary file 2. List of ts alleles that increased mHtt103QP toxicity in SGA screen.

• Supplementary file 3. List of *S. cerevisiae* strains and plasmids. A. List of *S. cerevisiae* strains. B. List of plasmids

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
