## [Decision Letter]

Thank you for submitting your work entitled "Spatial sequestration and detoxification of Huntingtin by the ribosome quality control complex" for consideration by *eLife*. Your article has been favorably evaluated by Naama Barkai (Senior editor) and three reviewers, one of whom, Andrew Dillin, is a member of our Board of Reviewing Editors.

The reviewers have discussed the reviews with one another and the Reviewing editor has drafted this decision to help you prepare a revised submission.

In this manuscript the authors investigate factors that influence the formation and toxicity of inclusion bodies (IB) formed upon expression of mutated Huntingtin protein (mHtt) with polyglutamine expansions in yeast. They identified components of the ribosome quality control complex (RQC), namely Ltn1, a ribosome-associated E3-ubiquitin ligase, together with Tae2 and the Hfs1, as key factors that affect IB formation and toxicity of mHtt. Loss of Ltn1 and Tae2 or a misregulation of Hsf1-activity provoked the formation of multiple mHtt aggregates instead of one large IB and a reduction in cell growth. In addition, using super-resolution microscopy, the authors showed that multiple and disperse mHtt aggregates alter the actin cytoskeleton and retard endocytosis processes.

While all three reviewers are excited about the work and its implications for the field, there are three major points that need to be addressed:

1) The ubiquitination experiments need to be performed with His-ubiquitin pull-down analysis from RQC deficient cells expressing mHtt.

2) Does Ltn1 deficiency cause abnormal actin cytoskeletal structures and reduced endocytosis independently of mHtt expression? How does their model fit with published data of Hsf1 and cytoskeletal dynamics?

3) The role of *HSF1* in this process could be direct as proposed, but also could be very indirect. It is suggested that the authors suggest alternative possibilities in their proposed models.

Finally, the idea that ribosomes could stall on repeat mRNAs, such as PolyQ containing, to elicit the RQC pathway is very interesting. All three reviewers feel that a direct demonstration of this stalling event on PolyQ mRNA would solidify the model, but we also felt that this might be a technically challenging experiment due to the repeat nature of the mRNA and alignment of the profile reads onto the correct message. Can the authors comment on the feasibility of this type of experiment?

*Reviewer #1:*

The dynamics and toxicity of PolyQ containing proteins is fascinating and not well understood. Here, the authors compare two forms of polyQ containing proteins (103Q) that differ by an internal proline region. The toxic form, lacking the P-rich region, forms small aggregates and the less toxic form, containing the P-rich domain forms a single large inclusion body. Asking if the IB formation is an active process, the group has screened for mutants that convert the IB into smaller bodies. The group focuses on Ltn1 and Rqc1 mutations, both of which have been identified for their role in ribosome associated protein quality control. Conversion of the smaller aggregates corresponds with increased toxicity and loss of Tae2, a downstream effector of Ltn1/Rqc1 suppresses both aggregate formation and toxicity.

One of the most interesting points of this work is the ability to accurately control toxic aggregate formation independent of ubiquitination of Htt. This is also the weakest point of the manuscript and few minor experiments will shore up the results. These include ubiquitin pull down experiments and probing for levels of Htt in the different mutant backgrounds.

Another interesting and unexpected finding is that loss of Tae2 in the Ltn1 mutant background restores IB formation and reduces toxicity. The main question to understand is whether the IB formed in the case is identical to the IB formed in the wild type setting (i.e. it is not ubiquitinated). It would excellent if solubility of the IBs could be measured (FRAP analysis for example). Also, are the IBs found in the ltn1 tae2 mutant also wrapped in actin cables?

Finally, the role of Hsf1 in this process is not fully detailed or alternative possibilities suggested. One thought to explain the dose dependency observed here could involve the idea that Hsf1 is a repressor of ltn1 or rqc1 expression. The analysis with the transactivation domain mutation of Hsf1 cannot address this possibility.

*Reviewer #2:*

Summary:

In this interesting study the authors have identified components involved in inclusion body (IB) formation of polyglutamine expanded huntingtin fragments (mHtt) through a genome-wide screen in yeast. They find that the ribosomal quality control complex (RQC) proteins, Ltn1 and Rqc1, are associated with the spatial sequestration of mHtt proteins and proteotoxicity. RQC deficient cells no longer sequester mHtt into IBs and this is attributed to over-activation of the heat stress response.

General comments:

The authors have presented the surprising finding that RQC deficiency renders cells deficient in sequestering mHtt proteins into IBs. This effect correlates with toxicity. However, the current manuscript does not differentiate between a direct or indirect role of RQC components in IB formation. RQC deficient cells may have a general proteostasis impairment resulting in failure of the actin cytoskeleton to mediate IB formation. Such proteostasis impairment may not occur (or be less pronounced) in the *TAE2* deletion strain, consistent with the absence of *HSF1* activation in that strain (Brandman et al.). The finding that *HSF1* activation (or suppression) reduces IB formation independently of RQC deficiency could be merely correlative. These aspects should be discussed in more detail. Another important question relates to the interesting possibility that Ltn1 is directly involved in mHtt ubiquitination (also see below).

Specific comments:

1) Data quality of mHtt ubiquitination in Figure 2 should be improved. His-ubiquitin pull-down analysis from RQC deficient cells expressing mHtt could be used.

2) Biochemical analysis following cycloheximide chase or metabolic chase should be performed to determine the stability of mHtt in *ltn1* deletion cells.

3) Does Ltn1 deficiency cause abnormal actin cytoskeletal structures and reduced endocytosis independently of mHtt expression?

4) What is the cut-off criteria for the HCM-based screen in [Supplementary-material SD1-data] and [Supplementary-material SD2-data]?

*Reviewer #3:*

In this manuscript the authors investigate factors that influence the formation and toxicity of inclusion bodies (IB) formed upon expression of mutated Huntingtin protein (mHtt) with polyglutamine expansions in yeast. They identified components of the ribosome quality control complex (RQC), namely Ltn1, a ribosome-associated E3-ubiquitin ligase, together with Tae2 and the Hfs1, as key factors that affect IB formation and toxicity of mHtt. Loss of Ltn1 and Tae2 or a misregulation of Hsf1 activity provoked the formation of multiple mHtt aggregates instead of one large IB and a reduction in cell growth. In addition, using super-resolution microscopy, the authors showed that multiple and disperse mHtt aggregates alter the actin cytoskeleton and retard endocytosis processes.

This is a very elegant study and the findings reported in this manuscript are new and very exciting. However, there are a few points that the authors should address prior to publication.

1) Ltn1 and other RQC components associate with disassembled 60S subunits that carry stalled nascent chains which are ubiquitinated by Ltn1 and turned over by the proteasome (Bengtson & Joazeiro, 2010). In this manuscript it remains unclear whether Ltn1 and other components of the RQC contribute directly or indirectly to the change in aggregate morphology and size and the toxicity of mHtt. Different scenarios are possible that are not mutually exclusive:

Expression of mHtt itself may cause stalling of translation and the production of truncated mHtt versions may induce Hsf1. Both, Hsf1 activity and truncated mHtt polypeptides, may contribute to alter aggregate morphology and mHtt toxicity.

Alternatively, expression of mHtt may challenge proteostasis in a more general manner (e.g. chaperone activities, proteasome activities etc.) and generally enhance stalling events that are handled by the RQC and induce Hsf1 activity. Thus, is it merely an indirect effect through HSR activation, followed e.g. by an increase in molecular chaperones, that holds the mHtt aggregates in an oligomeric, toxic state?

Further experimental analysis could shed some light on this issue. Does synthesis of mHtt (103QP) cause an arrest in translation and thus activate the RQC? Do truncated mHtt products occur in the absence of Ltn1? Is Ltn1 perhaps recruited from the ribosome to the mHtt1 aggregates? Does expression of mHtt generally enhance stalling frequency? To address these points, the authors can investigate stalling of mHtt via Western Blotting by expressing an N-terminally tagged version of mHtt in the presence and absence of Ltn1. Moreover, they can test for the stalling of a model protein (e.g. by 12Ks) during mHtt expression and for the association of Ltn1 with mHtt aggregates, e.g. by co-immunoprecipitation or co-localization in vivo.

In this context, recent publications that reconstitute the ubiquitination pathway at the ribosome and solved crystal structures of Ltn1 associated with 60S-RNCs should be considered briefly in the discussion (Shao & Hegde, 2013; Shao et al., 2015; Lyumkis et al., 2014; Shen et al. 2015).

2) This point relates to data shown Figure 2:

Figure 2 shows the ubiquitination of mHtt (103QP). It is unclear whether ubiquitination of soluble or aggregated 103QP is detected in the blot. Is this really ubiquitination of full-length mHtt (103QP) or of a truncated mHtt arrest product? Why is the ubiquitin signal different in wt compared to *tae2∆*? Could this argue for truncated mHtt products that lack the cat tail in tae2 mutant cells (see Shen et al., Science 2015)?

Please also include an immunoblot monitoring Ltn1 expression levels, especially when expressed from a plasmid.

Figure 2 shows the stability of mHtt in wt and ltn1∆ cells and the authors conclude that mHtt was equally stable in both strains. Does the readout include soluble AND/OR aggregated mHtt species? The readout was done by GFP activity (fused to mHtt). Thus, it may mainly report on insoluble aggregates of mHtt which show also GFP activity and of course remain rather stable! Moreover, the mHtt portion might be degraded by the proteasome but not the folded GFP moiety.

This experiment has to be re-designed to look directly to mHtt in the soluble vs. aggregated fraction in the presence and absence of Ltn1.

---

## [Author Response]

*In this manuscript the authors investigate factors that influence the formation and toxicity of inclusion bodies (IB) formed upon expression of mutated Huntingtin protein (mHtt) with polyglutamine expansions in yeast. They identified components of the ribosome quality control complex (RQC), namely Ltn1, a ribosome-associated E3-ubiquitin ligase, together with Tae2 and the Hfs1, as key factors that affect IB formation and toxicity of mHtt. Loss of Ltn1 and Tae2 or a misregulation of Hsf -activity provoked the formation of multiple mHtt aggregates instead of one large IB and a reduction in cell growth. In addition, using super-resolution microscopy, the authors showed that multiple and disperse mHtt aggregates alter the actin cytoskeleton and retard endocytosis processes. While all three reviewers are excited about the work and its implications for the field, there are three major points that need to be addressed: 1) The ubiquitination experiments need to be performed with His-ubiquitin pull-down analysis from RQC deficient cells expressing mHtt.*

The pull-down experiment has been performed as suggested and the data further support our conclusions concerning mHtt103QP ubiquitination. Please see Figure 2 and Figure 2—figure supplement 1.

*2) Does Ltn1 deficiency cause abnormal actin cytoskeletal structures and reduced endocytosis independently of mHtt expression? How does their model fit with published data of Hsf1 and cytoskeletal dynamics?*

We show that Ltn1 deficiency does not by itself cause abnormal actin cytoskeletal structures (Figure 3—figure supplement 2) or reduce endocytosis (Figure 3, the “vector” controls) but is rather epistatic to mHtt103QP expression.

We have added a discussion on, and reference to, the previously demonstrated role of Hsf1 in affecting actin cytoskeletal dynamics.

*3) The role of HSF1 in this process could be direct as proposed, but also could be very indirect. It is suggested that the authors suggest alternative possibilities in their proposed models.*

We agree and have added a discussion on this as suggested.

*Finally, the idea that ribosomes could stall on repeat mRNAs, such as PolyQ containing, to elicit the RQC pathway is very interesting. All three reviewers feel that a direct demonstration of this stalling event on PolyQ mRNA would solidify the model, but we also felt that this might be a technically challenging experiment due to the repeat nature of the mRNA and alignment of the profile reads onto the correct message. Can the authors comment on the feasibility of this type of experiment?*

We agree with the reviewers on the technical difficulty of such experiments and even if possible, it would most likely take much too long to get working properly than would be possible for the revision time allotted. Note also that the data included show that normal-size mHtt103QP is ubiquitinated in an Ltn1-dependent manner suggesting that stalling and the generation of truncated mHtt104QP species is not an absolute requirement for Ub tagging of mHtt103QP.

*Reviewer #1: One of the most interesting points of this work is the ability to accurately control toxic aggregate formation independent of ubiquitination of Htt. This is also the weakest point of the manuscript and few minor experiments will shore up the results. These include ubiquitin pull down experiments and probing for levels of Htt in the different mutant backgrounds.*

We have now included this analysis and show that data in Figure 2—figure supplement 1. The data further supports our previous conclusion.

*Another interesting and unexpected finding is that loss of Tae2 in the Ltn1 mutant background restores IB formation and reduces toxicity. The main question to understand is whether the IB formed in the case is identical to the IB formed in the wild type setting (i.e. it is not ubiquitinated). It would excellent if solubility of the IBs could be measured (FRAP analysis for example). Also, are the IBs found in the ltn1 tae2 mutant also wrapped in actin cables?*

We have performed this FRAP experiment as suggested and the data is included in Figure 2—figure supplement 2. We found that the dynamics/exchange within the IBs in *ltn1Δtae2Δ* double mutant (and the other mutants analyzed) are similar to those of Wt cells and that the IBs/aggregates co-localize with dense actin structure (see Figure 2—figure supplement 5).

*Finally, the role of sf-1 in this process is not fully detailed or alternative possibilities suggested. One thought to explain the dose dependency observed here could involve the idea that hsf-1 is a repressor of lt.-1 or rqc1 expression. The analysis with the transactivation domain mutation of hsf-1 cannot address this possibility.* We agree that the exact mechanistic role of Hsf1 in RQC-dependent IB formation is not clear and could include indirect effects and/or effects on regulating the levels and or activity of the RQC system. We have added comments related to this in the text. That Hsf1 would act as a repressor of *LTN1* cannot fully explain the results, however, as the suppression of the *ltn1∆* phenotype by a *tae2* deletion (reducing Hsf1 activity) occurs in cells lacking Ltn1.

*Reviewer #2: General comments:*

*The authors have presented the surprising finding that RQC deficiency renders cells deficient in sequestering mHtt proteins into IBs. This effect correlates with toxicity. However, the current manuscript does not differentiate between a direct or indirect role of RQC components in IB formation. RQC deficient cells may have a general proteostasis impairment resulting in failure of the actin cytoskeleton to mediate IB formation. Such proteostasis impairment may not occur (or be less pronounced) in the TAE2 deletion strain, consistent with the absence of HSF1 activation in that strain (Brandman et al.). The finding that HSF1 activation (or suppression) reduces IB formation independently of RQC deficiency could be merely correlative. These aspects should be discussed in more detail. Another important question relates to the interesting possibility that Ltn1 is directly involved in mHtt ubiquitination (also see below). Specific comments:*

1) Data quality of mHtt ubiquitination in Figure 2 should be improved. His-ubiquitin pull-down analysis from RQC deficient cells expressing mHtt could be used.

His-Ub pull-down analysis has been performed and included as suggested; the data is included in Figure 2—figure supplement 1. The data supports the previous conclusions.

*2) Biochemical analysis following cycloheximide chase or metabolic chase should be performed to determine the stability of mHtt in LTN1 deletion cells.*

This has been included in Figure 2—figure supplement 4.

*3) Does Ltn1 deficiency cause abnormal actin cytoskeletal structures and reduced endocytosis independently of mHtt expression?*

We have added this analysis and found that Ltn1-deficiency alone does not affect actin cytoskeletal integrity. The effect of *ltn1∆* is epistatic with mHtt103QP. The data has been included in Figure 3—figure supplement 1.

*4) What is the cut-off criteria for the HCM-based screen in [Supplementary-material SD1-data] and [Supplementary-material SD2-data]?*

The cut-off for HCM-based screen ([Supplementary-material SD1-data]) is 3 times of the variation of Wt class 3 values. The cut-off for SGA ([Supplementary-material SD2-data]) is -0.5 in the score from the screen. We have added information on this in the Methods section.

*Reviewer #3: This is a very elegant study and the findings reported in this manuscript are new and very exciting. However, there are a few points that the authors should address prior to publication. 1) Ltn1 and other RQC components associate with disassembled 60S subunits that carry stalled nascent chains which are ubiquitinated by Ltn1 and turned over by the proteasome (Bengtson & Joazeiro, 2010). In this manuscript it remains unclear whether Ltn1 and other components of the RQC contribute directly or indirectly to the change in aggregate morphology and size and the toxicity of mHtt. Different scenarios are possible that are not mutually exclusive:*

*Expression of mHtt itself may cause stalling of translation and the production of truncated mHtt versions may induce Hsf1. Both, Hsf1 activity and truncated mHtt polypeptides, may contribute to alter aggregate morphology and mHtt toxicity.*

*Alternatively, expression of mHtt may challenge proteostasis in a more general manner (e.g. chaperone activities, proteasome activities etc.) and generally enhance stalling events that are handled by the RQC and induce Hsf1 activity. Thus, is it merely an indirect effect through HSR activation, followed e.g. by an increase in molecular chaperones, that holds the mHtt aggregates in an oligomeric, toxic state?*

We agree with the reviewer that all these scenarios are possible and the data cannot exclude any of them at this point. We have expanded the Discussion to acknowledge these possibilities.

in vivo

We tried Westerns with anti-FLAG antibody but did not see any *ltn1Δ*-specific band. In addition, even though we detected smaller bands, we cannot exclude the possibility of degradation products.

*In this context, recent publications that reconstitute the ubiquitination pathway at the ribosome and solved crystal structures of Ltn1 associated with 60S-RNCs should be considered briefly in the discussion (Shao & Hegde, 2013; Shao et al., 2015; Lyumkis et al., 2014; Shen et al. 2015).*

We have added text on this as suggested.

2) This point relates to data shown Figure 2: Figure 2 shows the ubiquitination of mHtt (103QP). It is unclear whether ubiquitination of soluble or aggregated 103QP is detected in the blot. Is this really ubiquitination of full-length mHtt (103QP) or of a truncated mHtt arrest product? Why is the ubiquitin signal different in wt compared to tae2∆? Could this argue for truncated mHtt products that lack the cat tail in tae2 mutant cells (see Shen et al., Science 2015)?

*Please also include an immunoblot monitoring Ltn1 expression levels, especially when expressed from a plasmid.*

His-Ub pulled-down experiments has been performed for whole protein lysate and the results support our findings in Figure 2. Also, the pulled-down product shows the same size in Wt and *tae2∆*.

Figure 2 shows the stability of mHtt in wt and ltn1∆ cells and the authors conclude that mHtt was equally stable in both strains. Does the readout include soluble AND/OR aggregated mHtt species? The readout was done by GFP activity (fused to mHtt). Thus, it may mainly report on insoluble aggregates of mHtt which show also GFP activity and of course remain rather stable! Moreover, the mHtt portion might be degraded by the proteasome but not the folded GFP moiety.

*This experiment has to be re-designed to look directly to mHtt in the soluble vs. aggregated fraction in the presence and absence of Ltn1.*

Western-based cycloheximide chase experiment has now been included for both soluble and aggregated mHtt. The data is included in Figure 3—figure supplement 4. The data show that mHtt (soluble and aggregated) is not more stable in the *ltn1* mutant and that the total levels of mHtt is, in fact, somewhat lower in the *ltn1* mutant. Thus, the toxicity observed in Ltn1-deficient cells is not linked to elevated levels/stability of the protein.